# Neuromodulatory Signaling in Chronic Pain Patients: A Narrative Review

**DOI:** 10.3390/cells14171320

**Published:** 2025-08-27

**Authors:** Giustino Varrassi, Matteo Luigi Giuseppe Leoni, Giacomo Farì, Ameen Abdulhasan Al-Alwany, Sarah Al-Sharie, Diego Fornasari

**Affiliations:** 1Department of Research, Fondazione Paolo Procacci, 00193 Roma, Italy; 2College of Medicine, University of Bagdad, Bagdad 10071, Iraq; ameen.a@comed.uobaghdad.edu.iq; 3Department of Medical and Surgical Sciences and Translational Medicine, Sapienza University of Roma, 00135 Roma, Italy; matteolg.leoni@gmail.com; 4Department of Experimental Medicine, University of Salento, 73100 Lecce, Italy; giacomo.fari@unisalento.it; 5Department of Medicine, Division of Allergy, Pulmonary, and Critical Care Medicine, Vanderbilt University Medical Center, Nashville, TN 37232, USA; sarahalsharie2000@gmail.com; 6Department of Medical Biotechnology and Translational Medicine, Università degli Studi di Milano, 20122 Milano, Italy; diego.fornasari@unimi.it

**Keywords:** chronic pain, neuromodulation, neurotransmitters, BDNF, glial cells, central sensitization, neuroinflammation, descending inhibition, fibromyalgia, dopamine

## Abstract

Chronic pain is a complex and persistent condition involving sustained nociceptive input, maladaptive neuroplastic changes, and neuroimmune interactions. Central to its pathophysiology is the dysregulation of neuromodulatory signaling pathways, including neurotransmitters (e.g., dopamine, serotonin, norepinephrine), neuropeptides (e.g., substance P, CGRP), and neurotrophic factors (e.g., BDNF), which modulate both central and peripheral sensitization mechanisms. In disorders such as fibromyalgia, altered monoaminergic transmission has been implicated in the attenuation of descending inhibitory control, thereby enhancing pain perception and reducing responsiveness to conventional therapies. Concurrently, neuroinflammation, driven by glial cell activation and cytokine release, further exacerbates neuronal excitability and reinforces maladaptive signaling loops. Recent technological advances, including transcriptomic profiling, functional neuroimaging, and single-cell RNA sequencing, have provided new insights into patient-specific patterns of neuromodulatory dysfunction, highlighting potential biomarkers for disease stratification and therapeutic targeting. These developments support the hypothesis that dysregulated neuromodulatory circuits not only underlie diverse chronic pain phenotypes but may also serve as intervention points for precision medicine. This narrative review synthesizes current evidence on the roles of neuromodulatory systems in chronic pain, focusing on synaptic plasticity, nociceptor sensitization, and neuroimmune crosstalk. By integrating preclinical findings with clinical observations, we propose a mechanistic framework for understanding pain chronification and guiding future therapeutic strategies. Harnessing neuromodulatory targets, whether pharmacologically or via neuromodulation technologies, could offer more personalized and effective approaches to chronic pain management.

## 1. Introduction

Chronic pain is increasingly recognized as a complex, multidimensional disorder affecting an estimated 20% of the global population and representing a leading cause of long-term disability and socioeconomic burden worldwide [1,2]. Rather than being merely a prolonged manifestation of acute nociceptive signaling, chronic pain is now conceptualized as a distinct pathological condition involving maladaptive changes across multiple systems, including the nervous, endocrine, and immune systems [3]. This transformation is underpinned by sustained peripheral and central sensitization, disrupted endogenous pain modulation, emotional dysregulation, and long-term functional neuroplasticity in both cortical and subcortical structures [4,5].

One of the key mechanisms implicated in the chronification of pain is the dysfunction of endogenous neuromodulatory systems. Neuromodulators such as dopamine, serotonin, norepinephrine, glutamate, neuropeptides (e.g., substance P, CGRP), and neurotrophins, most notably Brain-Derived Neurotrophic Factor (BDNF), play essential roles in regulating synaptic plasticity, nociceptive thresholds, and descending inhibitory control [6,7]. These modulators interact with both neuronal and glial networks, coordinating the excitability of nociceptive pathways that span the peripheral nervous system (PNS), spinal cord dorsal horn, and brain regions including the periaqueductal gray (PAG), rostral ventromedial medulla (RVM), thalamus, and anterior cingulate cortex (ACC) [8].

Chronic pain conditions such as fibromyalgia, chronic migraine, irritable bowel syndrome, and various forms of neuropathic pain (e.g., diabetic polyneuropathy, postherpetic neuralgia, chemotherapy-induced neuropathy) have consistently demonstrated dysregulation in these neuromodulatory systems. For instance, impaired serotonergic and noradrenergic signaling has been associated with the attenuation of descending inhibitory control from the brainstem, contributing to the phenomenon of conditioned pain modulation (CPM) failure observed in many chronic pain patients [9,10]. Simultaneously, alterations in dopaminergic function (particularly in mesolimbic circuits) have been linked to diminished reward processing, increased affective pain burden, and decreased responsiveness to opioid analgesia [11,12].

Additionally, BDNF, a critical mediator of activity-dependent plasticity, has emerged as a key factor in the transition from acute to chronic pain. Increased levels of BDNF in the cerebrospinal fluid and serum of chronic pain patients correlate with enhanced N-methyl-D-aspartate (NMDA) receptor function, reduced GABAergic inhibition, and persistent hyperexcitability of dorsal horn neurons [13]. Furthermore, neuroimmune crosstalk via activated microglia and astrocytes releases cytokines and chemokines that reinforce central sensitization and perpetuate chronic pain states [14].

This narrative review explores the pathophysiological role of neuromodulatory signaling in chronic pain, with a focused analysis of neurotransmitter imbalances, maladaptive synaptic remodeling, and glial-neuronal interactions. Drawing on recent insights from molecular neuroscience, functional imaging, and transcriptomics, we aim to establish a comprehensive mechanistic framework that may inform biomarker discovery and the development of targeted, mechanism-based therapeutic interventions for chronic pain.

## 2. Materials and Methods

This narrative review was conducted in accordance with the SANRA (Scale for the Assessment of Narrative Review Articles) guidelines, which provide a validated framework to enhance the methodological rigor, transparency, and quality of non-systematic scientific literature [15]. The review was designed to provide a broad yet critical synthesis of the most relevant evidence concerning neuromodulatory mechanisms in chronic pain. A structured and comprehensive literature search was performed using major biomedical databases: PubMed, Web of Science (WoS), and Scopus. The search strategy incorporated both controlled vocabulary and free-text terms, including the keywords “chronic pain,” “neuromodulation,” “dopamine,” “serotonin,” “norepinephrine,” “brain-derived neurotrophic factor (BDNF),” “substance P,” “calcitonin gene-related peptide (CGRP),” “glial activation,” “neuroinflammation,” “descending pain modulation,” and “central sensitization.” These terms were used individually and in various Boolean combinations to maximize the retrieval of relevant publications.

The literature screening process prioritized peer-reviewed articles published in English between January 2015 and June 2025. Eligible publications included original research studies, systematic reviews, and meta-analyses with direct relevance to the neuromodulatory dimensions of chronic pain, encompassing both clinical and preclinical investigations. Preference was given to studies employing high-resolution neuroimaging, transcriptomic analyses, or advanced molecular techniques that elucidated the interaction between neurotransmitter systems, neuroimmune signaling, and pain chronification.

Studies were excluded if they focused exclusively on acute pain mechanisms, lacked a clear mechanistic component, or were published in languages other than English. Additional exclusion criteria involved low methodological quality or irrelevance to the thematic scope of this review. After de-duplication and screening, the references were appraised based on relevance, methodological robustness, citation impact, and contribution to current understanding.

The final selection represents a curated body of literature integrating molecular neuroscience, translational pain research, and clinical neurobiology. This approach ensures a well-rounded, evidence-informed synthesis that supports the development of a mechanistic framework for interpreting neuromodulatory dysfunction in chronic pain and identifying future avenues for therapeutic innovation.

## 3. Results

The literature search yielded a total of 1074 records: 420 from PubMed, 318 from Scopus, and 336 from WoS. After the removal of duplicate entries, 823 unique articles remained for screening. Titles and abstracts were then assessed for relevance, leading to the exclusion of 637 records that did not pertain to neuromodulatory mechanisms or chronic pain pathophysiology.

A total of 186 full-text articles were retrieved and evaluated against the predefined inclusion and exclusion criteria. Of these, 102 articles were excluded for reasons including: exclusive focus on acute pain models, peripheral nociceptive mechanisms without central involvement, publication in languages other than English, or insufficient methodological rigor (e.g., case reports or non-peer-reviewed commentary).

Ultimately, 84 articles met the inclusion criteria and were selected for inclusion in the narrative synthesis. These comprised 46 original research studies (clinical and preclinical), 22 systematic reviews or meta-analyses, and 16 high-impact narrative reviews published in reputable neuroscience, pain medicine, or molecular biology journals between 2015 and 2025. Figure 1 presents the PRISMA-style diagram detailing the study selection process. This rigorous curation ensures a comprehensive and focused synthesis of contemporary evidence on neuromodulatory signaling in chronic pain.

The selected publications were divided into 6 groups, by topic: Dopaminergic Signaling and Pain Regulation, Serotonergic and Noradrenergic Pathways, Brain-Derived Neurotrophic Factor (BDNF), Neuropeptides: Substance P and CGRP, Glial Cells and Neuroimmune Crosstalk, Transcriptomics and Functional Neuroimaging. Below, they are discussed extensively.

### 3.1. Dopaminergic Signaling and Pain Regulation

Dopaminergic signaling plays a critical role in the modulation of pain perception, primarily through its influence on mesolimbic, mesocortical, and nigrostriatal circuits as well as its involvement in descending inhibitory pain pathways [16,17,18,19,20,21,22,23,24]. The mesolimbic dopamine system, encompassing projections from the ventral tegmental area (VTA) to the nucleus accumbens (NAc), is traditionally associated with reward processing and motivation, yet is now increasingly recognized as a central node in the regulation of pain affect and relief expectancy [25,26,27]. Disruption of dopaminergic neurotransmission in this system has been consistently implicated in the pathogenesis of chronic pain syndromes [28].

Neuroimaging studies employing positron emission tomography (PET) have demonstrated reduced availability of dopamine D2/D3 receptors in key regions of the striatum in patients with fibromyalgia and complex regional pain syndrome (CRPS) [29,30]. This dopaminergic hypofunction correlates with increased pain intensity, heightened affective distress, and diminished pain inhibition, suggesting a direct link between D2 receptor activity and endogenous analgesic capacity. Functional MRI investigations further support these findings, revealing hypoactivation of the NAc and medial prefrontal cortex (mPFC) during anticipation of pain relief in fibromyalgia and other chronic pain conditions, pointing to a dysfunctional reward-related pain modulation system [11,31].

Preclinical studies in rodent models have substantiated the analgesic role of dopaminergic signaling, particularly through the activation of D2-like receptors. Systemic or spinal administration of D2 receptor agonists (e.g., quinpirole) has been shown to attenuate mechanical allodynia and thermal hyperalgesia in models of neuropathic and inflammatory pain [32]. In contrast, pharmacological blockade or genetic deletion of D2 receptors exacerbates pain behaviors, underscoring the inhibitory function of this receptor class in spinal nociceptive processing [33]. These effects appear to be mediated through presynaptic inhibition of nociceptive input and suppression of spinal neuron excitability in laminae I-II of the dorsal horn [34].

In addition to direct nociceptive modulation, impaired dopaminergic function also contributes to the emotional and cognitive burden of chronic pain. Reduced dopamine transmission in the mesocorticolimbic system is associated with anhedonia, reduced motivation, and depressive symptoms-features that are prevalent in patients with chronic pain and may exacerbate the experience of suffering [35]. The interplay between reward circuitry dysfunction and chronic pain highlights a shared neurobiological substrate that may partially explain the limited efficacy of traditional analgesics in these populations [36].

Emerging therapeutic strategies targeting the dopaminergic system (such as dopaminergic agonists, modulators of dopamine reuptake, and non-invasive brain stimulation targeting dopamine-rich regions) are under investigation and may offer novel approaches for patients with refractory pain syndromes, particularly those with prominent affective or cognitive symptoms [37,38]. Figure 2 summarizes the key anatomical pathways, experimental findings, and therapeutic targets associated with dopaminergic dysfunction in chronic pain.

### 3.2. Serotonergic and Noradrenergic Pathways

Descending pain modulation represents a critical endogenous mechanism for the inhibition of nociceptive signaling and plays a fundamental role in protecting the body from overwhelming or maladaptive pain responses [39]. Central to this system are brainstem structures such as the periaqueductal gray (PAG) and the rostral ventromedial medulla (RVM), which project to the dorsal horn of the spinal cord and modulate nociceptive transmission through both inhibitory and facilitatory pathways [40,41]. Among the key neurochemical substrates of descending inhibitory control are the monoamines serotonin (5-hydroxytryptamine, 5-HT) and norepinephrine (NE), which are released from descending projections originating in the brainstem, including the nucleus raphe magnus and locus coeruleus, respectively [42].

In healthy individuals, the monoaminergic system exerts a powerful antinociceptive influence through the activation of descending pathways that inhibit spinal nociceptive neurons via α2-adrenergic and 5-HT receptors, particularly 5-HT1A, 5-HT1B, subtypes [43]. However, in patients with chronic pain conditions such as fibromyalgia, diabetic neuropathy, and chronic low back pain, dysregulation of monoaminergic signaling has been consistently reported. Functional deficits in descending inhibitory control are frequently demonstrated using conditioned pain modulation (CPM) paradigms, which assess the integrity of diffuse noxious inhibitory control (DNIC) systems. Impaired CPM has been observed in numerous chronic pain populations and is closely associated with lower serum or cerebrospinal fluid levels of serotonin and norepinephrine, as well as with reduced responsiveness to analgesic stimuli [9,44].

This pathophysiological insight is strongly supported by clinical pharmacological data. Serotonin-norepinephrine reuptake inhibitors (SNRIs), including duloxetine and milnacipran, have demonstrated significant efficacy in treating various chronic pain syndromes, independent of their antidepressant properties [45]. These agents act by increasing the synaptic availability of 5-HT and NE in the central nervous system, thereby augmenting the descending inhibitory tone and restoring deficient antinociceptive pathways. Randomized controlled trials (RCTs) and meta-analyses have shown that duloxetine significantly reduces pain intensity and improves functional outcomes in patients with fibromyalgia, painful diabetic peripheral neuropathy, and chronic musculoskeletal pain, with numbers needed to treat (NNT) ranging from 5 to 7 for ≥30% pain reduction [46,47].

Moreover, the efficacy of SNRIs appears to correlate with the baseline functionality of descending inhibition, as demonstrated in mechanistic studies where greater improvements in pain scores were observed in individuals with initially impaired CPM responses [48]. These findings suggest that monoaminergic reuptake inhibition exerts its analgesic effect primarily by reinstating deficient top-down inhibitory control, rather than through direct modulation of peripheral nociceptive input.

Recent neuroimaging studies further support the involvement of monoaminergic circuits in pain modulation. Altered connectivity between the PAG, anterior cingulate cortex (ACC), and other pain-related brain regions has been documented in chronic pain patients, often in parallel with diminished serotonergic function assessed via molecular imaging techniques such as PET [49]. These alterations may reflect both structural and functional reorganization resulting from prolonged nociceptive input and impaired neurochemical feedback.

Taken together, current evidence underscores the critical role of serotonin and norepinephrine in maintaining endogenous pain inhibition. Their dysfunction contributes significantly to the pathophysiology of chronic pain and presents a validated therapeutic target for pharmacological intervention using SNRIs and related agents. Continued exploration of monoaminergic circuits may yield more refined and mechanism-specific approaches to the treatment of chronic pain syndromes.

### 3.3. Brain-Derived Neurotrophic Factor (BDNF)

Brain-Derived Neurotrophic Factor (BDNF), a member of the neurotrophin family, has emerged as a pivotal modulator of central sensitization and chronic pain pathophysiology [50]. BDNF is broadly expressed throughout the central and peripheral nervous systems and exerts its biological effects primarily through high-affinity binding to the tropomyosin receptor kinase B (TrkB). Under physiological conditions, BDNF plays a critical role in neuronal survival, differentiation, and synaptic plasticity [51]. However, in the context of chronic pain, aberrant expression and signaling of BDNF contribute to pathological neuronal excitability and maladaptive synaptic remodeling, particularly within the spinal dorsal horn and supraspinal nociceptive circuits [52].

Elevated levels of BDNF have been consistently detected in the cerebrospinal fluid (CSF), serum, and peripheral blood mononuclear cells (PBMCs) of patients suffering from fibromyalgia, chronic low-back pain, migraine, and other persistent pain conditions [53,54]. These elevated levels are believed to reflect an upregulation of activity-dependent BDNF synthesis and release from primary afferent fibers and activated microglia, triggered by sustained noxious stimulation and inflammatory signaling. Notably, BDNF is transported anterogradely from dorsal root ganglion (DRG) neurons to central terminals in the dorsal horn, where it modulates excitatory neurotransmission [55].

One of the principal mechanisms by which BDNF promotes central sensitization is through potentiation of N-methyl-D-aspartate (NMDA) receptor function. BDNF-TrkB signaling enhances NMDA receptor phosphorylation and trafficking to the synaptic membrane, thereby increasing calcium influx in postsynaptic neurons and promoting excitatory synaptic strength [56]. This effect contributes to the formation of long-term potentiation (LTP)-like changes in nociceptive pathways, a cellular correlate of chronic pain memory [57]. Concurrently, BDNF has been shown to downregulate the expression and activity of the potassium-chloride co-transporter 2 (KCC2), which is critical for maintaining chloride homeostasis in dorsal horn neurons [58]. This downregulation leads to a shift in the chloride gradient, rendering GABAergic and glycinergic inhibitory neurotransmission less effective or even excitatory. The resultant GABAergic disinhibition further exacerbates dorsal horn hyperexcitability and impairs the spinal gating of nociceptive input [59].

In preclinical models, intrathecal administration of BDNF induces mechanical allodynia and thermal hyperalgesia, mimicking features of central sensitization seen in chronic pain patients [60]. Conversely, blockade of BDNF or TrkB signaling, using neutralizing antibodies or small molecule inhibitors, has been shown to reduce pain behaviors and restore inhibitory tone in animal models of neuropathic and inflammatory pain [61].

Emerging data also implicate BDNF in supraspinal plasticity, particularly within pain-relevant areas such as the anterior cingulate cortex (ACC), insula, and thalamus. Altered BDNF signaling in these regions may contribute to the affective and cognitive symptoms of chronic pain, including anxiety, depression, and impaired emotional regulation [62].

In summary, BDNF is a central mediator of synaptic plasticity and neural circuit dysregulation in chronic pain. Its actions on NMDA receptor facilitation, KCC2 downregulation, and GABAergic disinhibition support the transition from acute to chronic pain states and represent promising targets for the development of mechanism-specific therapeutic interventions. Figure 3 illustrates the multifaceted role of BDNF in promoting central sensitization and neuronal hyperexcitability through both spinal and supraspinal mechanisms, highlighting its contribution to chronic pain pathophysiology.

### 3.4. Neuropeptides: Substance P and CGRP

Substance P and calcitonin gene-related peptide (CGRP) are two key neuropeptides critically involved in the pathogenesis of chronic pain through their roles in neurogenic inflammation, peripheral sensitization, and nociceptive signal amplification [63]. These peptides are co-released from the peripheral and central terminals of small-diameter primary afferent neurons, particularly C-fibers, in response to noxious stimuli. They serve as neuromodulators and inflammatory mediators, promoting a cascade of events that potentiate pain transmission and contribute to the transition from acute to chronic pain states [64,65].

Substance P, encoded by the TAC1 gene, is a member of the tachykinin family that exerts its biological effects primarily through the neurokinin-1 (NK1) receptor [66]. Upon release, it induces plasma extravasation, vasodilation, and mast cell degranulation, thereby initiating and sustaining neurogenic inflammation. Moreover, Substance P enhances the excitability of peripheral nociceptors and facilitates neurotransmission at spinal cord synapses, particularly in lamina I and II of the dorsal horn [67]. Some findings also suggest its involvement in central sensitization by upregulating excitatory synaptic responses and downregulating GABAergic inhibition [68].

CGRP, a 37-amino acid peptide derived from alternative splicing of the CALCA gene, is widely distributed in the trigeminovascular system and dorsal root ganglia [69]. It is one of the most potent endogenous vasodilators and a key player in inflammatory pain processes. CGRP acts through the CGRP receptor complex (composed of calcitonin receptor-like receptor and receptor activity-modifying protein 1) to induce smooth muscle relaxation, promote vasodilation, and enhance immune cell recruitment via cytokine release [70]. In peripheral tissues, CGRP increases vascular permeability and sensitizes nociceptors by lowering their activation threshold, contributing to hyperalgesia and allodynia [71].

Elevated levels of Substance P and CGRP have been consistently observed in plasma and cerebrospinal fluid (CSF) of patients with chronic pain conditions, particularly fibromyalgia, chronic migraine, and complex regional pain syndrome (CRPS) [72,73,74]. In fibromyalgia, higher serum levels of CGRP and Substance P have been associated with greater pain severity, fatigue, and sleep disturbances, suggesting a role in the maintenance of symptom clusters [72]. In migraine, CGRP is implicated in the initiation and maintenance of headache episodes through its vasodilatory and pro-inflammatory effects within the trigeminovascular system. Notably, CGRP levels have been shown to rise during migraine attacks and normalize following treatment with triptans [75].

The translational relevance of CGRP has been strongly validated by the clinical success of monoclonal antibodies targeting either the CGRP ligand (e.g., fremanezumab, galcanezumab, eptinezumab) or its receptor (e.g., erenumab). These agents have demonstrated efficacy in reducing migraine frequency, intensity, and duration in both episodic and chronic migraine populations, with favorable safety profiles and prolonged action due to their long half-lives [76,77,78]. Their approval by regulatory agencies such as the FDA and EMA underscores the therapeutic potential of targeting CGRP in neurogenic pain conditions.

Despite the established role of CGRP in migraine, emerging studies are now exploring its broader involvement in other pain conditions such as osteoarthritis, fibromyalgia, and visceral hypersensitivity syndromes, potentially expanding the scope of anti-CGRP therapies [70]. Similarly, NK1 receptor antagonists, although less successful in initial clinical trials for pain, are under renewed investigation in combination therapies and for targeting specific phenotypes of centralized pain [79].

In conclusion, Substance P and CGRP are pivotal neuromodulators in pain pathophysiology. Their ability to mediate neurogenic inflammation, vascular responses, and peripheral sensitization not only elucidates fundamental mechanisms of chronic pain but also offers validated and promising targets for mechanism-specific therapies. The role of Substance P and CGRP in neurogenic inflammation, peripheral sensitization, and central sensitization, including their molecular mechanisms and downstream effects in chronic pain states, is summarized in Figure 4.

### 3.5. Glial Cells and Neuroimmune Crosstalk

Chronic pain is now widely recognized as a disorder involving not only neurons but also non-neuronal cells, particularly glial elements such as microglia and astrocytes [80]. These glial cells, which are traditionally understood as supportive components of the central nervous system (CNS), have emerged as key modulators of neuroinflammation and neuronal excitability in both spinal and supraspinal pain circuits. In the context of persistent nociceptive input, glial activation becomes sustained, contributing to the establishment and maintenance of central sensitization, a hallmark of chronic pain [14,81].

Microglia, the resident immune cells of the CNS, respond rapidly to pathological stimuli such as nerve injury, inflammation, or tissue damage. Upon activation, microglia undergo phenotypic transformation and release a range of pro-inflammatory mediators, including tumor necrosis factor-alpha (TNF-α), interleukin-1 beta (IL-1β), interleukin-6 (IL-6), and various chemokines such as CCL2 and CXCL1. These signaling molecules act on nearby neurons to enhance excitability, lower pain thresholds, and disrupt inhibitory neurotransmission, particularly in lamina I and II of the spinal dorsal horn [82,83]. Astrocytes, although activated later than microglia, contribute to the maintenance phase of chronic pain through the release of similar pro-inflammatory cytokines and modulation of glutamate uptake, which further intensifies neuronal hyperexcitability [84].

Recent mechanistic studies have elucidated specific pathways involved in microglial activation in chronic pain states [85]. Toll-like receptor 4 (TLR4), a pattern recognition receptor expressed on microglia, is upregulated following peripheral nerve injury and drives downstream MyD88/NF-κB signaling, leading to cytokine production and sustained inflammation [86]. Similarly, P2X7 purinergic receptors, activated by extracellular ATP, are crucial for microglial proliferation and inflammasome activation, facilitating the release of IL-1β via NLRP3 signaling pathways [87]. Another pivotal axis involves the CX3CL1-CX3CR1 (fractalkine) interaction between neurons and microglia, which promotes microglial chemotaxis, cytokine expression, and direct modulation of synaptic function [88].

Emerging evidence also indicates that glial activation patterns exhibit sex-specific differences, reflecting distinct neuroimmune mechanisms underlying chronic pain [89]. In preclinical models, male rodents exhibit microglia-dependent pain hypersensitivity following nerve injury, whereas females appear to rely more heavily on adaptive immune cells, particularly T lymphocytes, for the development of central sensitization [90]. These sex-dependent pathways may be mediated by differential expression of immune receptors, hormonal modulation, and epigenetic regulation of inflammatory gene transcription [89].

Sex differences extend beyond glial mechanisms to encompass all major neuromodulatory systems discussed in this review. Female patients with fibromyalgia show greater serotonergic dysfunction compared to males, potentially explaining the higher prevalence and severity of the condition in women [91]. Similarly, BDNF responses to pain stimuli exhibit sexual dimorphism, with females showing greater activity-dependent release and more pronounced central sensitization following nerve injury [92]. These differences have therapeutic implications: women may respond better to serotonergic interventions, while men may benefit more from dopaminergic approaches [93]. CGRP levels and receptor sensitivity also vary by sex, with estrogen modulating CGRP expression and contributing to menstrual migraine patterns [94]. Understanding these sex-specific mechanisms is crucial for developing personalized pain management strategies. Chronic pain’s female predominance, notably in fibromyalgia, reflects hormonal, neurochemical, and neuroimmune interactions. Estrogen fluctuations lower nociceptive thresholds and weaken endogenous inhibition during low-estrogen phases, while progesterone modulates GABAergic tone and descending control. Testosterone offers male protection via anti-inflammatory effects and enhanced inhibition. These hormonal influences, combined with female-biased CGRP expression, T-cell sensitization, and heightened HPA axis reactivity, create sex-specific pain phenotypes and higher chronification risk in women, highlighting the need for sex-specific research and therapies. Such findings challenge the historical neuron-centric paradigm of chronic pain and underscore the importance of neuroimmune interactions. Importantly, pharmacological strategies targeting glial activation have shown promise in preclinical studies. For instance, TLR4 antagonists, P2X7 inhibitors, and glial metabolic disruptors (e.g., propentofylline, minocycline) have attenuated pain behaviors in various models of neuropathic and inflammatory pain [95]. However, translating these interventions to clinical settings remains a challenge, with issues of specificity, sex-responsiveness, and timing of intervention being major considerations.

Neuroinflammation occurs in many neurological disorders, but its progression to chronic pain may depend on factors such as its location within nociceptive pathways, distinct cytokine patterns (IL-1β, TNF-α, BDNF), the persistence of glial activation, and the integrity of descending inhibitory control. These represent possible hypotheses rather than established mechanisms, as clear evidence on how these variables interact to produce pain remains lacking. This uncertainty may explain why similar lesions cause pain in some patients but not others, and why ALS or Alzheimer’s inflammation seldom results in pain, highlighting the need for further research to guide translation of anti-inflammatory strategies into clinical practice.

Astrocytes, although activated later than microglia, contribute to the maintenance phase of chronic pain through the release of similar pro-inflammatory cytokines and modulation of glutamate uptake. This dysfunction is characterized by the downregulation of glutamate transporters EAAT1/GLAST and EAAT2/GLT-1, as well as impaired glutamine synthetase activity, which together reduce glutamate clearance and promote excitotoxicity, thereby enhancing neuronal hyperexcitability and sustaining central sensitization.

A comprehensive overview of the role of glial cells—particularly microglia and astrocytes—in neuroimmune crosstalk, illustrating their activation mechanisms, cytokine release, and contribution to neuronal hyperexcitability and central sensitization in chronic pain in reported in Figure 5.

In conclusion, microglia and astrocytes contribute fundamentally to the pathogenesis of chronic pain through sustained activation and the release of pro-inflammatory mediators. Their engagement in neuron-glia and glia-glia communication networks establishes a self-perpetuating cycle of sensitization and neuroinflammation. Understanding the molecular and sex-specific intricacies of glial signaling holds the potential to unlock novel, mechanism-based treatments for chronic pain syndromes.

### 3.6. Transcriptomics and Functional Neuroimaging

Transcriptomic and neuroimaging studies have significantly advanced the understanding of the molecular and systems-level alterations underlying chronic pain disorders [96]. In recent years, transcriptomic profiling of human dorsal root ganglion (DRG) neurons and human peripheral blood mononuclear cells (PBMCs) has revealed distinct patterns of gene expression associated with pain chronification [97,98]. These analyses have identified upregulation of genes related to neuroinflammation (e.g., TNF, IL1B, CCL2), neuronal excitability (e.g., SCN9A, TRPV1, CACNA1H), and neuromodulatory signaling (e.g., BDNF, TAC1, OPRM1) in patients with various chronic pain phenotypes, including fibromyalgia, neuropathic pain, and chronic low back pain. Moreover, alterations in the expression of genes involved in immune-neural interactions and synaptic plasticity suggest that peripheral immune dysregulation may contribute to central nervous system (CNS) sensitization through feed-forward signaling loops [99].

Single-cell RNA sequencing (scRNA-seq) of both human post-mortem and rodent DRG tissue has further refined the classification of sensory neuron subtypes and revealed subtype-specific transcriptional signatures in models of neuropathic and inflammatory pain [100]. In mouse models, selective enrichment of injury-induced markers (e.g., ATF3, SPRR1A) in nociceptive subsets, alongside dynamic changes in purinergic, peptidergic, and ionotropic receptor gene expression that modulate nociceptor responsiveness [101,102]. Human DRG studies have confirmed similar epigenetic modifications, such as altered histone acetylation and DNA methylation, have been implicated in the persistent reprogramming of sensory neurons in both rodent models and human tissue samples, contributing to long-term changes in pain processing and treatment responsiveness [103].

Complementing molecular studies, neuroimaging techniques such as functional magnetic resonance imaging (fMRI) and positron emission tomography (PET) have delineated reproducible alterations in brain network connectivity among individuals with chronic pain [104]. In patients with fibromyalgia and related conditions, disruptions have been consistently observed in the salience network (including the anterior insula and anterior cingulate cortex), the default mode network (DMN; encompassing the medial prefrontal cortex and posterior cingulate cortex), and the executive control network (e.g., dorsolateral prefrontal cortex) [11,105]. These network abnormalities in patients correlate with clinical pain severity, cognitive dysfunction (e.g., “fibrofog”), and affective symptoms such as anxiety and depression [106].

Moreover, PET imaging studies in patients using radioligands targeting glial markers (e.g., TSPO) and neurotransmitter systems (e.g., dopaminergic and opioid receptors) have revealed reduced receptor binding potential and increased neuroinflammatory signals in regions implicated in pain modulation, reward, and emotional regulation [107,108]. These findings collectively suggest that chronic pain is not solely a peripheral disorder but involves systemic neuroimmune interactions and large-scale brain network dysregulation, reinforcing the need for mechanism-based, personalized approaches to diagnosis and treatment.

### 3.7. Translational Challenges and Clinical Perspectives

Despite promising preclinical findings, the translation of neuromodulatory targets to clinical practice has faced significant challenges [109]. Several high-profile failures illustrate the complexity of chronic pain translation: NK1 receptor antagonists, despite robust preclinical efficacy, failed in multiple Phase III trials for various pain conditions [110]. Similarly, glial modulators such as propentofylline and ibudilast have shown limited clinical efficacy despite strong preclinical rationale [111]. These translational failures highlight critical gaps between rodent models and human chronic pain. Species differences in immune responses, particularly the greater reliance on adaptive immunity in humans compared to rodents, may explain why microglial-targeted therapies have struggled clinically [112]. Additionally, the heterogeneity of chronic pain phenotypes in humans contrasts sharply with the homogeneous models used in preclinical research. Most preclinical studies evaluate pain responses over days to weeks, whereas human chronic pain develops over months to years with complex temporal dynamics that cannot be captured in short-lived animal models [113]. Furthermore, approximately 80% of preclinical pain studies use exclusively male rodents, overlooking critical sex differences in pain mechanisms—female mice show T-cell-mediated pain sensitization versus microglial-mediated mechanisms in males [90], while estrogen modulates CGRP expression and receptor sensitivity in ways not captured by male-only studies [94]. Behavioral assessment limitations compound these issues: rodent pain tests rely primarily on reflexive withdrawal responses that poorly correlate with spontaneous pain—the primary complaint in human chronic pain—and cannot capture pain qualities (burning, shooting, aching) critical for diagnosis and treatment selection [114].

Recent failures have prompted a shift toward mechanism-based patient stratification. The success of CGRP antibodies in migraine, where patients were selected based on specific pathophysiological criteria rather than broad diagnostic categories, provides a template for future development [115]. Similarly, the identification of responder phenotypes for duloxetine based on conditioned pain modulation status suggests that functional biomarkers may improve therapeutic targeting [48]. However, while molecular and neuroimaging findings provide mechanistic insights, their correlation with patient-reported pain experiences remains complex and often weak [116]. Studies examining the relationship between biomarkers and pain intensity have shown modest correlations (r = 0.3–0.5), suggesting that biological measures capture only part of the pain experience [117]. Patient-reported outcomes in chronic pain trials reveal important disconnects between objective measures and subjective experience. In fibromyalgia studies, improvements in inflammatory markers or brain connectivity do not always translate to meaningful clinical improvement (defined as ≥30% pain reduction), underscoring the multidimensional nature of chronic pain that encompasses sensory, affective, and cognitive components not fully captured by biological markers [8]. Quality of life measures, including sleep quality, physical function, and emotional well-being, often show stronger predictive value for treatment response than neuroimaging or molecular markers [9]. The consistent failure of promising preclinical targets in clinical trials—with success rates below 10% for analgesic drug development—emphasizes that successful therapies must address both neurobiological mechanisms validated through patient stratification and the lived experience of pain captured through patient-reported outcome measures [10,114].

## 4. Discussion

The results of this review underscore the complex interplay between neuromodulatory, neuroimmune, and synaptic signaling systems in the pathogenesis and persistence of chronic pain. Dopaminergic dysfunction emerged as a central mechanism, particularly involving the mesolimbic and mesocortical circuits traditionally linked to reward and affective processing. Reduced dopamine D2/D3 receptor availability in the striatum, as shown in PET studies of patients with fibromyalgia and CRPS, correlates with pain severity and affective distress [29,30]. Functional MRI studies further reveal hypoactivation of the nucleus accumbens and medial prefrontal cortex during pain relief anticipation, indicating disrupted reward-pain integration [11,31]. These alterations may underlie both sensory amplification and the emotional burden of chronic pain [35,36].

Simultaneously, impairments in descending monoaminergic control, particularly serotonin and norepinephrine pathways projecting from the periaqueductal gray (PAG) and rostral ventromedial medulla (RVM), contribute to reduced endogenous pain inhibition [39,40,41,42]. Dysregulation of these systems is evident in conditions such as fibromyalgia and diabetic neuropathy and is associated with impaired conditioned pain modulation (CPM) [9,44,118]. The efficacy of serotonin-norepinephrine reuptake inhibitors (SNRIs), such as duloxetine and milnacipran, in restoring descending inhibition validates this mechanism and demonstrates therapeutic relevance [45,46,47].

BDNF, a key neurotrophin, contributes to maladaptive synaptic plasticity and central sensitization through TrkB receptor-mediated potentiation of NMDA receptor activity and downregulation of KCC2 transporters [56,57,58,59]. These effects culminate in GABAergic disinhibition and sustained spinal hyperexcitability, facilitating long-term potentiation (LTP)-like changes that encode persistent pain [57]. Elevated BDNF levels in CSF and serum of chronic pain patients further support its role in pain maintenance [53,54].

At the periphery, neuropeptides such as Substance P and CGRP mediate neurogenic inflammation and peripheral sensitization [63,64,65]. CGRP, in particular, has translational importance; monoclonal antibodies targeting the CGRP ligand or receptor (e.g., erenumab, fremanezumab) have shown efficacy in reducing migraine frequency and intensity [76,77,78]. Elevated serum levels of Substance P and CGRP in fibromyalgia and migraine correlate with symptom clusters including fatigue and pain intensity [72,75].

Glial activation represents an emerging therapeutic target. Microglia and astrocytes, when chronically activated, release pro-inflammatory mediators (e.g., TNF-α, IL-1β) that amplify nociceptive signaling [82,83,84]. Activation via TLR4, P2X7, and fractalkine receptors initiates sustained neuroimmune cascades, with sex-dependent variations observed, microglial dominance in males and T-cell-driven signaling in females [86,87,88,89,90]. Pharmacological inhibition of these glial pathways (e.g., minocycline, propentofylline) shows promise in preclinical models, though translation to clinical care remains challenging [95].

Finally, transcriptomic studies and neuroimaging converge on the concept of chronic pain as a systems-level disorder. Altered gene expression in DRG neurons and PBMCs (e.g., SCN9A, TRPV1, BDNF) reflects peripheral sensitization and neuroimmune activation [97,98,99]. Simultaneously, fMRI and PET studies consistently reveal disrupted functional connectivity in the salience, default mode, and executive control networks, changes that correlate with cognitive dysfunction and affective symptoms [104,105,106]. These insights underscore the importance of integrating molecular and network-level diagnostics to guide mechanism-based treatments in chronic pain management.

While the mechanistic evidence appears robust, several methodological concerns warrant consideration. Many neuroimaging studies suffer from small sample sizes (typically *n* = 15–30 per group) and may be confounded by medication effects, comorbid depression, and chronic pain duration [116]. The reproducibility of transcriptomic findings across different platforms and populations remains limited, with effect sizes often modest and clinical correlations weak [117]. Preclinical evidence, while mechanistically informative, may overestimate therapeutic potential due to publication bias favoring positive results and the use of young, healthy, male rodents that poorly represent the human chronic pain population [113]. The failure of numerous promising targets in clinical trials (e.g., NK1 antagonists, NMDA modulators) underscores the need for more rigorous preclinical validation and better animal models that incorporate aging, comorbidities, and sex differences [114].

Despite promising preclinical findings, the translation of neuromodulatory targets to clinical practice has faced significant challenges [109]. Several high-profile failures illustrate the complexity of chronic pain translation: NK1 receptor antagonists, despite robust preclinical efficacy, failed in multiple Phase III trials for various pain conditions [110]. Similarly, glial modulators such as propentofylline and ibudilast have shown limited clinical efficacy despite strong preclinical rationale [111]. These translational failures highlight critical gaps between rodent models and human chronic pain. Species differences in immune responses, particularly the greater reliance on adaptive immunity in humans compared to rodents, may explain why microglial-targeted therapies have struggled clinically [112]. Additionally, the heterogeneity of chronic pain phenotypes in humans contrasts sharply with the homogeneous models used in preclinical research. Recent failures have prompted a shift toward mechanism-based patient stratification. The success of CGRP antibodies in migraine, where patients were selected based on specific pathophysiological criteria rather than broad diagnostic categories, provides a template for future development [115]. Similarly, the identification of responder phenotypes for duloxetine based on conditioned pain modulation status suggests that functional biomarkers may improve therapeutic targeting [48].

### Limitations

This review is subject to several limitations. First, as a narrative synthesis, it lacks the rigor and reproducibility of systematic and meta-analytical approaches, which may limit the generalizability of its conclusions. Second, many of the cited mechanistic studies rely on preclinical models that may not fully capture the heterogeneity and complexity of human chronic pain. Translational discrepancies between rodent and human findings, particularly in glial and sex-dependent mechanisms, remain unresolved. Third, while advanced neuroimaging and transcriptomic analyses provide valuable insights, they often involve small sample sizes and may suffer from confounding due to comorbid mood disorders, medication use, or age-related changes. Moreover, the mechanisms underlying why neuroinflammation leads to pain in some conditions but not others remain poorly understood, limiting our ability to predict which patients will develop chronic pain after neural injury or inflammation. This gap highlights the urgent need for targeted studies to clarify the determinants of inflammation–pain specificity and improve patient stratification. Finally, the multifactorial nature of chronic pain means that no single pathway or biomarker can serve as a universal target, highlighting the need for personalized, multimodal approaches. Additionally, the predominant reliance on animal models that poorly represent human chronic pain populations is a critical translational barrier. This mismatch may explain the high failure rate of promising preclinical targets in clinical trials. Future studies integrating longitudinal omics, imaging, and clinical phenotyping are necessary to validate the reviewed mechanisms and translate them into precision therapies.

## 5. Conclusions

Chronic pain is a multifaceted disorder underpinned by complex interactions among neuromodulatory dysfunction, neuroimmune activation, synaptic plasticity, and large-scale brain network reorganization. Dysregulation of dopaminergic, monoaminergic, neurotrophic, and peptidergic pathways contributes to both sensory and affective dimensions of pain, while glial activation and transcriptomic alterations reinforce central sensitization. Integrating molecular and systems-level findings offers a more comprehensive understanding of chronic pain pathophysiology and highlights the potential for targeted, mechanism-based interventions. Future research should prioritize personalized approaches that align therapeutic strategies with individual neurobiological profiles.

## Figures and Tables

**Figure 1 cells-14-01320-f001:**
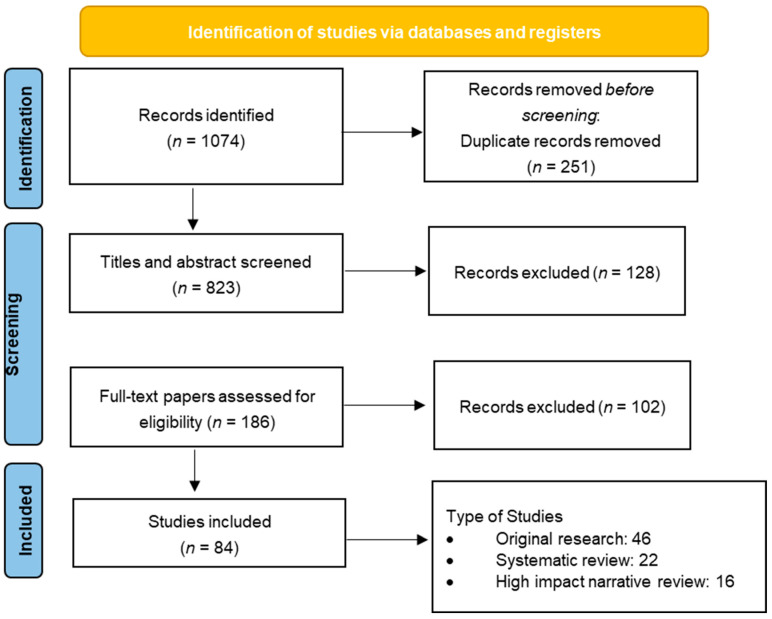
PRISMA flow diagram outlining the selection process of articles included in the review.

**Figure 2 cells-14-01320-f002:**
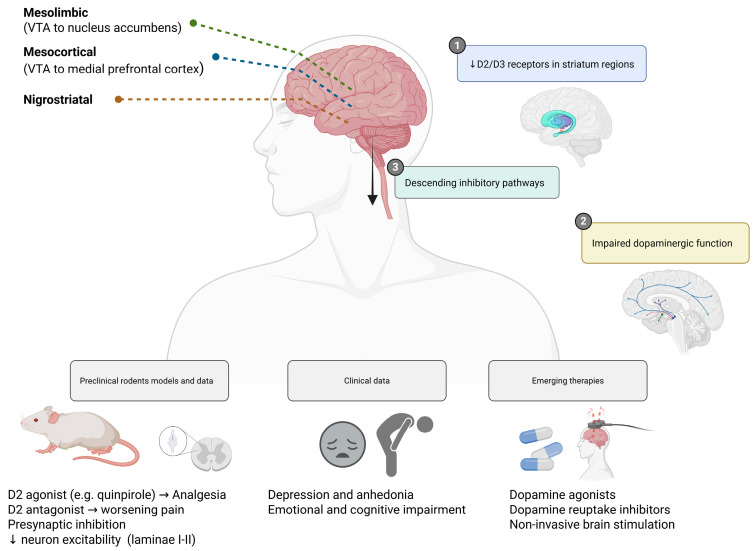
Dopaminergic modulation of pain and emerging therapeutic strategies. Dopaminergic circuits—including mesolimbic (VTA to nucleus accumbens), mesocortical (VTA to medial prefrontal cortex), and nigrostriatal pathways—play a pivotal role in pain modulation and affective processing. In chronic pain states, reduced D2/D3 receptor availability and impaired dopamine signaling are linked to heightened pain perception, anhedonia, and emotional distress. Preclinical models showed that D2 agonists confer analgesic effects, while D2 antagonists worsen pain responses. Emerging therapies aim to restore dopaminergic tone through dopamine agonists, reuptake inhibitors, or non-invasive brain stimulation techniques. Created in https://BioRender.com.

**Figure 3 cells-14-01320-f003:**
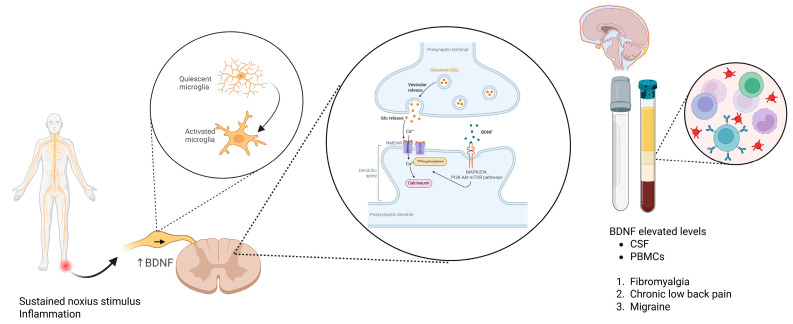
Mechanisms of BDNF-induced central sensitization in chronic pain. BDNF released from inflamed tissue, microglia, and dorsal root ganglion (DRG) neurons binds to TrkB receptors on spinal dorsal horn neurons. This leads to NMDA receptor phosphorylation, increased calcium influx, and downregulation of the potassium-chloride cotransporter KCC2, resulting in impaired inhibitory neurotransmission and enhanced neuronal excitability. Elevated BDNF levels have been detected in CSF and PBMCs of patients with fibromyalgia, chronic low-back pain, and migraine, supporting its role as a biomarker and therapeutic target. Created in https://BioRender.com.

**Figure 4 cells-14-01320-f004:**
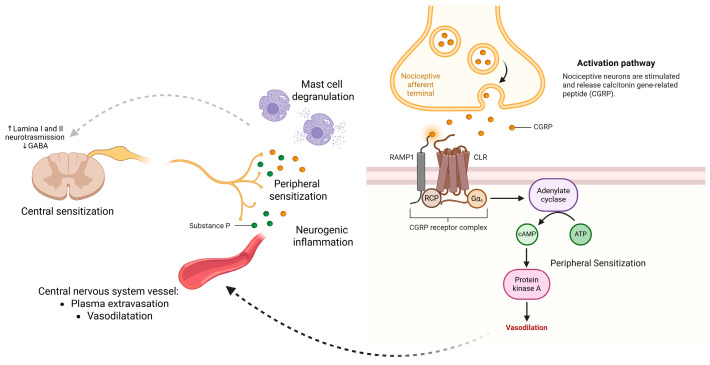
Mechanisms of Substance P and CGRP in chronic pain. Substance P contributes to neurogenic inflammation by promoting plasma extravasation, vasodilation, and mast cell degranulation, leading to peripheral and central sensitization via enhanced excitatory neurotransmission and GABAergic inhibition in lamina I–II of the dorsal horn. CGRP, released from trigeminal and sensory neurons, activates the CGRP receptor complex (CLR, RAMP1, RCP), leading to increased cAMP, activation of protein kinase A, and vasodilation—amplifying peripheral sensitization. Created in https://BioRender.com.

**Figure 5 cells-14-01320-f005:**
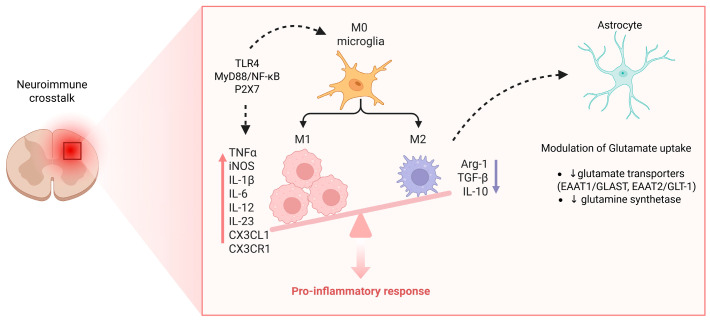
Microglia activation, astrocytic dysfunction, and neuroimmune crosstalk in chronic pain. Resting (M0) microglia transition into a pro-inflammatory (M1) phenotype via TLR4–MyD88–NF-κB and P2X7 signaling, releasing mediators such as TNF-α, IL-1β, IL-6, IL-12, IL-23, CX3CL1, and iNOS that amplify neuronal excitability and sustain central sensitization. In contrast, the M2 phenotype (producing Arg-1, IL-10, TGF-β) contributes to inflammation resolution but is suppressed in chronic pain. Astrocytic dysfunction, characterized by downregulation of glutamate transporters (EAAT1/GLAST, EAAT2/GLT-1) and glutamine synthetase, impairs glutamate uptake and clearance, thereby enhancing excitotoxicity and reinforcing the pro-inflammatory response. Together, these microglial and astrocytic changes create a self-perpetuating cycle of neuroinflammation driving chronic pain. Created in https://BioRender.com.

## Data Availability

The data supporting the conclusions of this narrative review are derived from published literature sources cited throughout the manuscript. All references and their corresponding data are publicly available through their respective journals and databases. Additional information regarding the literature search strategy, study selection criteria, or data extraction processes are available from the corresponding author upon reasonable request.

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
