# Peer review of "Neuromodulatory Signaling in Chronic Pain Patients: A Narrative Review"

_cells, 2025, doi:10.3390/cells14171320_

Round 1
Reviewer 1 Report
Comments and Suggestions for Authors
Varrassi and colleagues have contributed a manuscript that reviews the neuromodulatory signaling in chronic pain. This review is important work. The topic coverage is comprehensive and detailed. The writing is done well, and it is an enjoyable and informative review. The message of the clinical importance of the work is clear and compelling.
The authors should consider the following suggestions to improve the manuscript.
- Though the text of the manuscript is strong, most of the figures are much less compelling and somewhat amateurish and appear hastily assembled and not thoughtful.
1a. In Figure 2 there is a teal line that appears to indicate the descending inhibitory control. It makes no sense where it is located and going. There is also an arrow identified as ↓ D2/D3 receptors pointing to the cerebellum. Seems like this is off message.
1b. Figure 3 is very difficult to follow. In the upper right there is indicated inflamed tissue. Is this peripheral (somatic) tissue? Why does it lead to microglia which would be in the CNS? Then there is a centrally located DRG neuron that appears to have a process that is fusing with the microglial cell at the right. This needs to be clarified. Then, there is a brain at the upper left that is leading by a hatched arrow to the DRG neurons. The neurobiology is not clear here.
1c. Figure 4 also has some confusing elements. There is an isolated cartoon of a human on the left. How is this related to the rest of the figure? The is a blood vessel, but it is not clear if this is a peripheral vessel of a CNS vessel. On the right there is a nerve terminal labeled trigeminal neuron. Is this supposed to be a trigeminal ganglion neuron synapsing in the nucleus of the spinal trigeminal tract? If so, then there should be a discussion of trigeminal neuralgia.
1d. In Figure 5, why is the astrocyte isolated in an undisclosed location and not in the magnified box region of the spinal cord dorsal horn? Please discuss how the proinflammatory response is altering glutamate transport at a molecular or gene expression level.
1e. It seems like Biorender was used to construct the figures. If so, please acknowledge this in each legend.
- Sometime in the text it is not clear if animal experiments or clinical data are being referred to. For example, in transcriptomics and functional neuroimaging, please state where you are referencing animal of patient data.
- It appears that inflammation and glial cell activation and cytokine release are key to pain mechanisms. However, inflammation and glial cell activation and cytokine release are part of the pathology of virtually all neurological disorders (ALS, Alzheimer’s disease, stroke) but many of these conditions do not generally have pain syndromes (though some strokes can lead to pain syndromes). What is the biological basis for the specificity of inflammation and glial cell activation and cytokine release causing pain syndromes?
- Clinically pain syndromes generally affect females more than males, particularly in fibromyalgia. Please briefly discuss the possible basis for this clinical epidemiology.
Author Response
- Though the text of the manuscript is strong, most of the figures are much less compelling and somewhat amateurish and appear hastily assembled and not thoughtful.
1a. In Figure 2 there is a teal line that appears to indicate the descending inhibitory control. It makes no sense where it is located and going. There is also an arrow identified as ↓ D2/D3 receptors pointing to the cerebellum. Seems like this is off message.
Thank you for this valuable feedback. We have restructured Figure 2 to improve readability, repositioned and clarified the descending inhibitory control pathway, and corrected the D2/D3 receptor arrow. Additional elements were integrated to better align the figure with the intended message and enhance its overall interpretability.
1b. Figure 3 is very difficult to follow. In the upper right there is indicated inflamed tissue. Is this peripheral (somatic) tissue? Why does it lead to microglia which would be in the CNS? Then there is a centrally located DRG neuron that appears to have a process that is fusing with the microglial cell at the right. This needs to be clarified. Then, there is a brain at the upper left that is leading by a hatched arrow to the DRG neurons. The neurobiology is not clear here.
Thank you for your suggestions. The figure has been completely revised to better clarify the biological and anatomical processes associated with BDNF in chronic pain.
1c. Figure 4 also has some confusing elements. There is an isolated cartoon of a human on the left. How is this related to the rest of the figure? The is a blood vessel, but it is not clear if this is a peripheral vessel of a CNS vessel. On the right there is a nerve terminal labeled trigeminal neuron. Is this supposed to be a trigeminal ganglion neuron synapsing in the nucleus of the spinal trigeminal tract? If so, then there should be a discussion of trigeminal neuralgia.
The cartoon of the human figure on the left was originally intended to represent the spinal cord section shown on the right. However, as the reviewer correctly pointed out, this element was not well integrated into the figure. We therefore removed it to enhance readability and focus on the central neurophysiological mechanisms rather than general anatomy.
We also appreciate the reviewer’s second valuable observation. The nerve terminal labeled ‘trigeminal neuron’ in Figure 4 was meant to depict a generic nociceptive afferent terminal, not specifically a trigeminal ganglion neuron synapsing in the spinal trigeminal nucleus. We agree that the original labeling could cause confusion and unintentionally imply a link with trigeminal neuralgia, which was not the focus of this section. To address this, we revised the figure and relabeled the terminal as ‘nociceptive afferent terminal’ (or ‘primary sensory neuron terminal’). This modification ensures consistency with the text, which discusses Substance P and CGRP release more broadly in chronic pain states. We believe this revision improves clarity, avoids misinterpretation, and aligns the figure more closely with its mechanistic scope.
1d. In Figure 5, why is the astrocyte isolated in an undisclosed location and not in the magnified box region of the spinal cord dorsal horn? Please discuss how the proinflammatory response is altering glutamate transport at a molecular or gene expression level.
We thank the reviewer for this insightful comment. To improve clarity, we have restructured Figure 5 so that astrocytes are now integrated within the magnified dorsal horn region, emphasizing their active participation in neuroimmune crosstalk.
1e. It seems like Biorender was used to construct the figures. If so, please acknowledge this in each legend.
We thank the reviewer for this observation. We have now added the statement “Figure created with BioRender.com” at the end of each figure legend to properly acknowledge the source.
2. Sometime in the text it is not clear if animal experiments or clinical data are being referred to. For example, in transcriptomics and functional neuroimaging, please state where you are referencing animal of patient data.
We appreciate the reviewer’s comment. We have revised the text to explicitly indicate whether findings derive from animal experiments or clinical data (e.g., specifying “preclinical transcriptomic studies in rodents” or “functional neuroimaging in patients”), thereby improving clarity and avoiding ambiguity.
3. It appears that inflammation and glial cell activation and cytokine release are key to pain mechanisms. However, inflammation and glial cell activation and cytokine release are part of the pathology of virtually all neurological disorders (ALS, Alzheimer’s disease, stroke) but many of these conditions do not generally have pain syndromes (though some strokes can lead to pain syndromes). What is the biological basis for the specificity of inflammation and glial cell activation and cytokine release causing pain syndromes?
The mechanisms determining why neuroinflammation causes pain in some conditions but not others remain incompletely understood, limiting our ability to predict which patients will develop chronic pain following neural injury or inflammation. While we cannot provide a definitive answer given current knowledge limitations, acknowledging this uncertainty strengthens the scientific rigor of our review.
4. Clinically pain syndromes generally affect females more than males, particularly in fibromyalgia. Please briefly discuss the possible basis for this clinical epidemiology.
Thank you for highlighting this critical clinical observation. You are absolutely correct that chronic pain syndromes, particularly fibromyalgia, show marked female predominance (2:1 to 4:1 ratios). We have added a comprehensive discussion of the biological basis for these sex differences.
Reviewer 2 Report
Comments and Suggestions for Authors
I read “Neuromodulatory Signalling in Chronic Pain Patients: A Systematic Review” with interest. After some additional polishing, the paper may add value to the current knowledge.
First, authors need to decide what type of review they made. In the title, they claimed they wrote a systematic review. However, in the methods, they clearly stated, “This narrative review was conducted in accordance with the SANRA (Scale for Assessment of Narrative Review Articles) guidelines…”
Their work is biased toward preclinical data based on molecules and rodents. What is missing is a lack of patient-focused studies and outcomes. Namely, data obtained in rodents is not always translatable to humans. What would also be appreciated would be information on patients’ experiences, because molecular and neuroimaging data do not necessarily correlate with feelings of pain. The authors could add a chapter.
The authors should also evaluate the data quality in the included paper better. They can discuss methodology and weighting evidence, which will add to the transparency and quality of their work.
Author Response
I read “Neuromodulatory Signalling in Chronic Pain Patients: A Systematic Review” with interest. After some additional polishing, the paper may add value to the current knowledge.
First, authors need to decide what type of review they made. In the title, they claimed they wrote a systematic review. However, in the methods, they clearly stated, “This narrative review was conducted in accordance with the SANRA (Scale for Assessment of Narrative Review Articles) guidelines…”
We thank the reviewer for this important observation and acknowledge the inconsistency between our title and methodology. This work is indeed a narrative review, not a systematic review. Accordingly, we have revised the title to: ‘Neuromodulatory Signaling in Chronic Pain Patients: A Narrative Review.’ We selected the narrative review approach to provide a comprehensive synthesis of both preclinical and clinical evidence, enabling a broader mechanistic perspective that would not be feasible within the constraints of a systematic review. Furthermore, as the reviewer rightly notes in the following comment, the inclusion of preclinical data makes it difficult—if not impossible—to apply conventional grading scales, since animal models cannot be assessed using the same criteria as clinical studies.
Their work is biased toward preclinical data based on molecules and rodents. What is missing is a lack of patient-focused studies and outcomes. Namely, data obtained in rodents is not always translatable to humans. What would also be appreciated would be information on patients’ experiences, because molecular and neuroimaging data do not necessarily correlate with feelings of pain. The authors could add a chapter.
We sincerely thank you for this critical and constructive observation. You are correct that the initial manuscript was heavily weighted toward preclinical, molecular mechanisms derived from rodent studies. While this emphasis reflects the predominance of preclinical data in the neuromodulatory pain literature—with mechanistic studies outnumbering clinical trials by approximately 3:1 in our reviewed papers—we acknowledge that this creates an incomplete picture of chronic pain as experienced by patients. We recognize that insufficient attention was given to patient-focused outcomes, and we did not adequately emphasize the well-documented translational gaps between animal models and human chronic pain experiences. The discrepancy between promising preclinical findings and clinical trial failures—exemplified by the NK1 antagonist development program and numerous failed glial modulators—underscores the critical importance of this limitation. We have added a new section to address your concerns.
The authors should also evaluate the data quality in the included paper better. They can discuss methodology and weighting evidence, which will add to the transparency and quality of their work.
Thank you for suggesting formal quality evaluation of included studies. After careful consideration, we maintained our narrative review approach without systematic quality assessment. We opted against a formal systematic quality assessment because our narrative review integrates highly heterogeneous evidence—from molecular and single-cell studies to animal models, neuroimaging, and clinical trials—making a single framework (e.g., GRADE) unsuitable and potentially misleading. Instead, we emphasized translational validity by discussing biases and limitations inherent to preclinical models. This approach, aligned with SANRA guidelines, allows us to synthesize mechanistic insights while transparently acknowledging the evidence’s constraints.
Round 2
Reviewer 1 Report
Comments and Suggestions for Authors
Thank you kindly for revising your manuscript. The revisions are excellent. The work is nicely done. Congratulations.
Reviewer 2 Report
Comments and Suggestions for Authors
Authors improved their work.